# *In vitro* evolution of an influenza broadly neutralizing antibody is modulated by hemagglutinin receptor specificity

Nicholas C. Wu[1], Geramie Grande[2], Hannah L. Turner[1], Andrew B. Ward[1], Jia Xie[2], Richard A. Lerner[2] & Ian A. Wilson[1,3]

The relatively recent discovery and characterization of human broadly neutralizing antibodies (bnAbs) against influenza virus provide valuable insights into antiviral and vaccine development. However, the factors that influence the evolution of high-affinity bnAbs remain elusive. We therefore explore the functional sequence space of bnAb C05, which targets the receptor-binding site (RBS) of influenza haemagglutinin (HA) via a long CDR H3. We combine saturation mutagenesis with yeast display to enrich for C05 variants of CDR H3 that bind to H1 and H3 HAs. The C05 variants evolve up to 20-fold higher affinity but increase specificity to each HA subtype used in the selection. Structural analysis reveals that the fine specificity is strongly influenced by a highly conserved substitution that regulates receptor binding in different subtypes. Overall, this study suggests that subtle natural variations in the HA RBS between subtypes and species may differentially influence the evolution of high-affinity bnAbs.

[1] Department of Integrative Structural and Computational Biology, The Scripps Research Institute, La Jolla, California 92037, USA. [2] Department of Chemistry, The Scripps Research Institute, La Jolla, California 92037, USA. [3] The Skaggs Institute for Chemical Biology, The Scripps Research Institute, La Jolla, California 92037, USA. Correspondence and requests for materials should be addressed to J.X. (email: jiaxie@scripps.edu) or to I.A.W. (email: wilson@scripps.edu).

Seasonal and pandemic influenza A viruses have a global impact on human health and the world economy. Despite decades of research, forecasting the genetic evolution of the virus remains extremely difficult due to rapid antigenic drift and antigenic shift[1]. Cross-species transmission further increases the unpredictability of influenza genetic dynamics in nature. Annual vaccination is the only available prophylactic measure, but its efficacy is far from perfect and can vary based on the accuracy of vaccine strain prediction[2]. In terms of therapeutics, there are currently two classes of anti-influenza drugs, namely neuraminidase inhibitors and M2 protein inhibitors. However, drug-resistant mutants for both inhibitor classes have emerged and are present in circulating strains[3,4], suggesting an urgent need for new therapeutic agents.

The relatively recent discovery of human influenza broadly neutralizing antibodies (bnAbs) against the stem region of haemagglutinin (HA)[5–14] has provided significant insights for antiviral and vaccine development. For example, stem-binding antibodies CR6261 (refs 5,6) and CR8020 (ref. 7) are in clinical trials (NCT02371668 and NCT01938352) as antivirals. In addition, small protein binders, that were computational designed[15,16] based on the epitope information of a stem-binding bnAb CR6261 (refs 5,6), provided in vivo protection from influenza challenge[17]. These stem-binding antibodies have also guided the development of immunogens that confer hetero-subtypic protection[18–20]. Recently, another class of influenza bnAbs that target the HA receptor-binding site (RBS) has been identified and characterized[21–31]. Each of these HA RBS-targeted bnAbs features a long hypervariable loop that inserts into the HA RBS. While some HA RBS-targeted bnAbs, including 8M2 (refs 26,27), CH65 (ref. 28) and 5J8 (refs 29,30), are subtype-specific due to the higher sequence variability in the RBS and its proximal regions as compared to the stem region, others, including C05 (ref. 21), S139/1 (refs 22,23), F045-092 (refs 24,25) and 2G1 (refs 26,27), display heterosubtypic activity. Extrapolating from the success in harnessing information from stem-binding bnAbs, these HA RBS-targeted bnAbs offer an unprecedented opportunity to develop new influenza antivirals and potentially a more universal vaccine against HA RBS.

The HA RBS is a shallow pocket in the globular HA head[32], and is framed by four structural elements: 130-loop, 150-loop, 190-helix and 220-loop, named after their positions on the HA primary sequence. Although a large portion of the HA RBS is extremely conserved across subtypes, some minor but very important sequence and structural variations exist that affect receptor-binding preferences and interaction mode among different subtypes and across species. For example, residues 190 and 225 (H3 numbering) are Asp in H1 human influenza strains, but Glu and Gly in most H2 and H3 human as well as avian influenza strains. The amino-acid sequences of the 220-loop also differ across subtypes and species, where residues 226 and 228 are generally Leu and Ser in human H2 and H3 subtypes, but Gln and Gly in human H1 and avian subtypes. However, it is unclear how these amino-acid variations shape the evolution of HA RBS-targeted bnAbs in sequential infection by the same or different subtypes.

Understanding the natural evolution of influenza bnAb is important towards development of a universal influenza vaccine. It is evidenced that original antigenic sin strongly influences the immune response against influenza virus[33,34]. For example, influenza vaccination efficacy is higher in people with no prior influenza vaccination history than in those with frequent influenza vaccination[35]. Immune history may also prevent influenza bnAbs from being elicited[36]. In addition, while many antibodies induced during the 2009 pandemic H1N1 influenza season exhibited broad neutralization activity against the HA stem[37], this response was not sustained in later years[36,38]. These observations indicate that our understanding of the evolution and maintenance of influenza bnAbs is far from complete.

Here we aimed to study the evolution of an HA RBS-targeted bnAb C05 (ref. 21) by examining its functional sequence space (that is, the sequence requirements for binding). C05 was originally discovered from a human donor with confirmed influenza virus exposure using phage display library technology[21]. C05 neutralizes strains from the pandemic subtypes H1, H2, H3, as well as an H9 virus, using primarily a long, single complementarity determining region loop (CDR H3)[21] that inserts into the RBS. Specifically, we performed saturation mutagenesis on the six-residue paratope region of C05 CDR H3 that interacts within the heart of the HA RBS. Subsequently, yeast display was used to independently enrich for C05 variants that bound to H1 HA (A/Solomon Islands/3/2006), and H3 HA (A/Perth/16/2009). Several C05 variants retained binding to H3 HA but lost affinity against H1 HA. We also identified C05 variants that increased binding to H1 HA, but not against H3 HA. Further analysis suggests that the amino-acid preference of the paratope region of interest becomes subtype-specific despite targeting the conserved RBS using affinity maturation in this selection process. To provide structural insights, we determined the crystal structure of a C05 Fab variant that was enriched to high frequency in selections both against H1 and H3 HAs in complex with the HA1 subunit of the HA trimer to 1.97 Å resolution. Together, our results unequivocally demonstrate that the HA subtype preference of the increased affinity C05 is, at least partly, attributed to the amino-acid identity at residue 190 in the HA RBS. In particular, Ser is favoured at position 100d of the C05 heavy chain when residue 190 of HA1 is an Asp and is disfavoured when residue 190 of HA is a Glu. This implies that even highly conservative substitutions that dictate HA receptor preference and mode of binding also modulate the amino-acid preference in the paratope of the antibody, which has important implications in the evolution of bnAbs to the RBS and in the development of more universal vaccines that target the RBS.

## Results

**Yeast display screening of C05 Fab functional variants.** In the past few years, a number of neutralizing antibodies that target the influenza HA receptor-binding site (HA RBS) have been identified[21–31]. Notably, one such antibody, C05, uses a single loop on heavy-chain complementarity-determining region 3 (CDR H3) to target the HA RBS (Supplementary Fig. 1a) and is able to neutralize several, although not all, strains within pandemic subtypes from both group 1 (H1, H2) and group 2 (H3) influenza viruses[21]. Six consecutive amino acids at the tip of C05 CDR H3 (positions 100a to 100f) directly contact the HA RBS, which is largely conserved across influenza subtypes (Supplementary Fig. 1b). This scaffold provides an opportunity to comprehensively examine the functional sequence space of the HA RBS-targeted residues in C05, which should provide valuable information in understanding the evolution of HA RBS-targeted broadly neutralizing antibodies (bnAb) and in antiviral development against the HA RBS.

To examine the functional sequence space of the HA RBS-targeted residues in C05, we constructed a mutant library and performed screening and selection by yeast display. The amino-acid sequence of the six residues in CDR H3 in wild-type (WT) C05 that most intimately interact with the RBS is $_{100a}$VVSAGW$_{100f}$. Saturation mutagenesis was applied to the first five amino acids (position 100a to 100e), whereas mutagenesis of the tryptophan at position 100f was restricted to the first row of the codon table, which included Phe, Leu, Ser, Tyr, Cys and Trp.

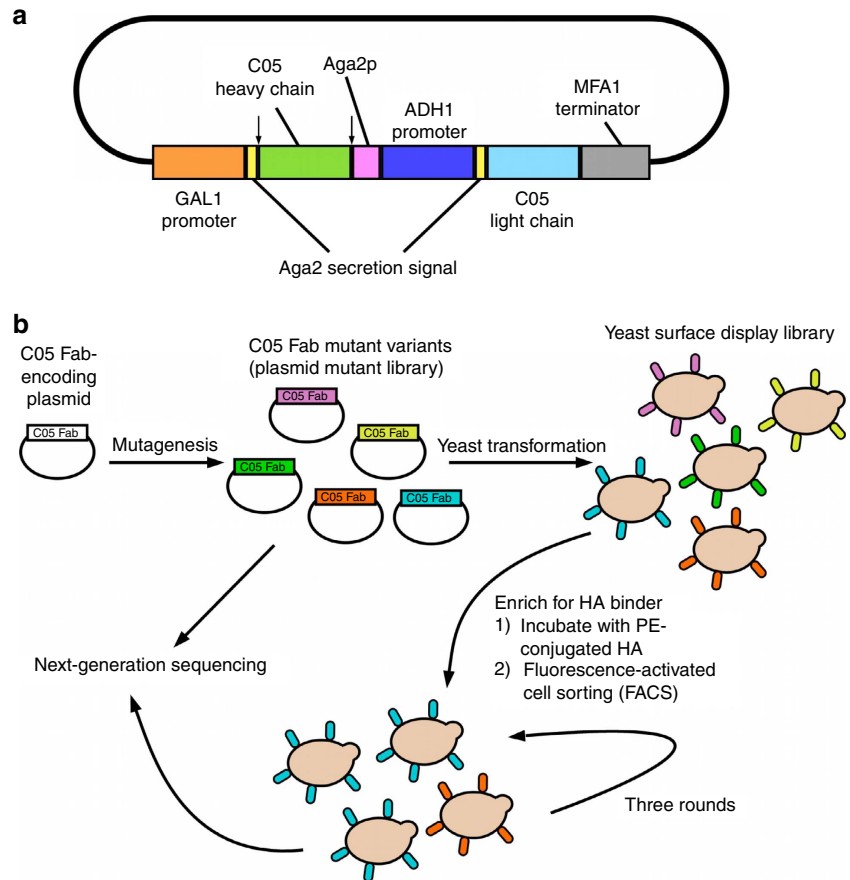

**Figure 1 | Schematic representation of the yeast display screening.** (**a**) Schematic representation of the dual promoter yeast expression plasmid encoding the C05 Fab is shown. The arrows indicate the locations of the SfiI sites employed for cloning. (**b**) Saturation mutagenesis was applied to the six amino acids of C05 CDR H3 that interact closely or are buried in the receptor-binding site of influenza hemagglutinin (HA). A C05 Fab plasmid mutant library, which consists of at least $10^8$ clones, was created. The plasmid mutant library was transformed into yeast strain EBY100 to generate a yeast surface display library. Different variants of C05 Fab were displayed on yeast cells. This yeast surface display library was then subjected to selection for HA-binding affinity. We used fluorescence-activated cell sorting (FACS) to enrich yeast cells that were able to interact with PE-conjugated influenza HA. The post-selection pool was then expanded and subjected to another round of selection. For each of H1, H3, and H5 HAs, three rounds of selection was performed. Variants that were able to bind to the HA would enrich in occurrence frequency throughout the screening process. Variants with higher affinity would enrich to a higher frequency. The plasmid mutant library and each of the post-selection mutant libraries were next-generation sequenced to monitor the frequency change of each variant.

The rationale of applying a more restricted mutagenesis scheme at position 100f as compared to other positions was because: (1) bulky hydrophobic residues were suggested to be critical for HA RBS-targeted neutralizing antibodies at this spatial location[31]; and (2) compared to saturation mutagenesis, this mutagenesis scheme would reduce the sequence diversity by more than threefold, which would increase the oversampling of each sequence variant during the screening process. The resultant mutant library has an amino-acid sequence diversity of $\sim 20$ million variants ($20^5 \times 6 = 19,200,000$), within the estimated throughput limit ($\sim 100$ million) for yeast display screening[39]. The sequence diversity of these six residues of interest in the mutant library should be higher than that generated by somatic hypermutation, which is mostly limited to substitutions resulting from single-nucleotide changes. Our pilot experiment indicated that C05 does not interact with HA in a single-chain variable fragment (scFv) format. Subsequently, the mutant library was constructed based on the fragment antigen-binding (Fab) format. Specifically, we developed a dual promoter yeast expression plasmid, which enabled expression of both the heavy and light chains of C05 Fab from a single plasmid (Fig. 1a).

This mutant library was then screened by yeast display[40] for affinity against an H1 HA (A/Solomon Islands/3/2006, also referred to as SI06 below), an H3 HA (A/Perth/16/2009), and an H5 HA (A/Vietnam/1203/2004; Fig. 1b). Wild-type (WT) C05 is able to bind to H1 and H3 HAs, but not H5 HA due to a steric clash outside of the paratope region of interest. Therefore, H5 HA was utilized as a negative control for this screening process. For each of H1, H3 and H5 HAs, the screening was composed of three rounds of selection. For each round, we used fluorescence-activated cell sorting (FACS) to enrich yeast cells that were able to interact with PE-labelled influenza HA. Variants with higher affinity would then be enriched throughout the screening process. As expected, enrichment of functional variants was observed in the selection against H1 HA and H3 HA, but not against H5 HA (Supplementary Fig. 2). Next-generation sequencing was performed after each round of selection to monitor the frequency of each variant within the mutant library. Next-generation sequencing revealed that WT C05 Fab from incomplete digestion of vector during library construction had a much higher occurrence frequency (1 in 100) than each variant in the input mutant library (1 in 19,200,000 on average). As a result, it was not surprising to observe that WT C05 Fab was

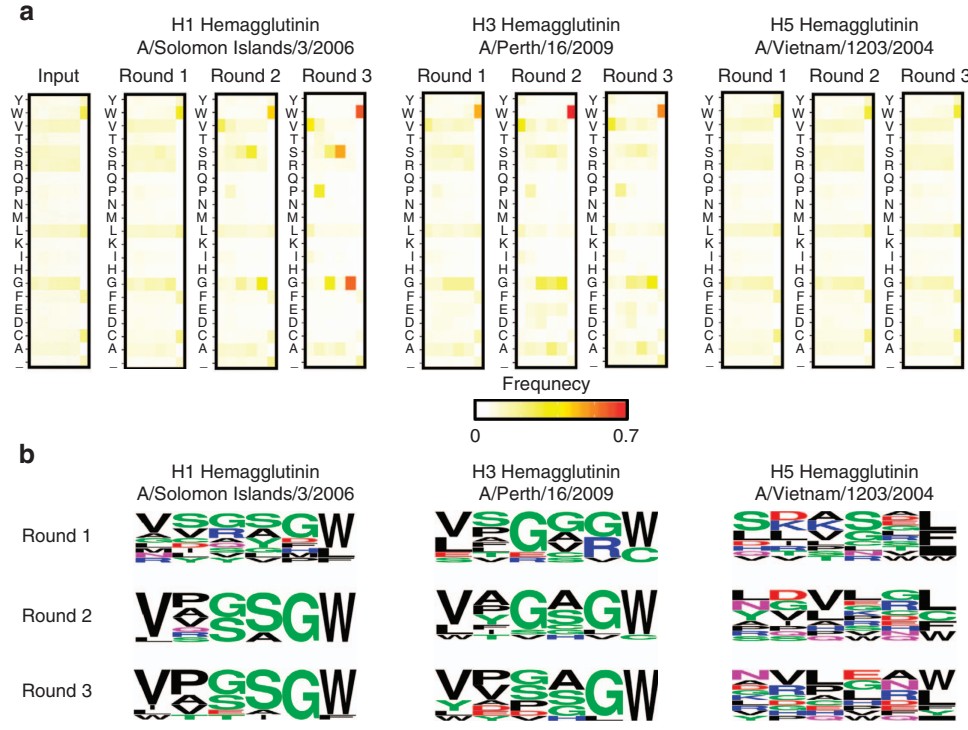

**Figure 2 | Monitoring the selection process by next-generation sequencing.** (**a**) The occurrence frequency of each amino acid at each of the 6 residues of interest is shown as a heatmap. (**b**) For each round of selection, a sequence logo was generated using the amino-acid sequences of the top 10 variants with the highest occurrence frequency.

enriched to >80% by round 3 against H1 HA and H3 HA (Supplementary Fig. 3). Due to the high occurrence frequency relative to all the variants in the mutant library, WT C05 Fab from incomplete digestion of vector was excluded from our downstream analysis (see Methods). Throughout the manuscript, the identity of a given variant is denoted by the amino-acid sequence of the six residues in CDR H3 that were probed by mutagenesis and selection (for example, WT C05 Fab is VVSAGW).

**CDR H3 amino-acid preference and top variants.** To understand the amino-acid preference at each position of the six amino-acid residues at the tip of the CDR H3 loop, the occurrence frequency of each amino acid at individual positions was examined (Fig. 2a). We observed that Trp at position 100f was significantly enriched throughout the selection against both H1 and H3 HAs. Enrichments were also observed for Val at position 100a, Pro at position 100b, and Gly at position 100e, and were more prevalent in the selection against H1 HA compared to H3 HA. In addition, Ser at position 100d was enriched in selection against H1 HA. The detection of amino-acid enrichment indicates the success of the selection against the H1 and H3 HAs. In contrast, no enrichment was observed in selection against H5 HA (negative control). We further analysed the enrichment of individual variants (Supplementary Fig. 4). The top 10 variants after three rounds of selection against H1 HA reached a frequency ranging from 1.3% to 11.5% (from highest to lowest frequency: VPGSGW, LPSSGW, VPGAGW, VASSGW, IPGSGW, VTGSGW, VATSGW, VVSSGW, VVGSGW and WPEIGF). The top 10 variants after three-round selection against H3 HA reached a frequency of ranging from 0.7% to 2.1% (from highest to lowest: VPGAGW, WYVHLW, LPGGGW, YDPGGW, VPGSGW, VVSSGW, VVSAGW, VVDSGW,

VASAGW and YEPAGW). The top 10 variants after three-round selection against H5 HA reached a frequency of ranging from 0.1% to 0.5% (from highest to lowest: YVHPQF, NPQEEL, RVLVRL, VVPEFW, NGCGRW, ARELAY, GHLHNW, DRPLAW, KCLWN_, LDAGDL, where '_' indicates a stop codon). A sequence logo was created for the top 10 variants from each round of selection (Fig. 2b). Overall, the results from next-generation sequencing suggest that the paratope region of interest in C05 can tolerate a number of amino-acid substitutions without abolishing affinity against HA, and indeed can increase affinity in some cases.

**HA subtype-dependent amino-acid preference of C05 Fab.** We aimed to experimentally validate the next-generation sequencing results from yeast display screening. We therefore compiled a list of 43 variants from the top 20 variants by occurrence frequency in each of the round 3 selections against H1 HA and against H3 HA, and the top eight variants by occurrence frequency in round 3 selection against H5 HA. Of note, the same five variants, one of which is WT, appeared in the top 20 variants of round 3 selections against H1 HA and H3 HA (VPGAGW, VPGSGW, VVGSGW, VVSSGW and VVSAGW). We were able to express 40 out of 43 candidates in HEK293T cells and measured their relative affinities against H1 HA and H3 HA using unpurified cell culture supernatants (see Methods, Fig. 3a). Variants with improved affinity against HA were observed. From the top 20 variants in the selection against H1 HA, five variants have increased affinity against H1 HA. For the top 20 variants in the selection against H3 HA, two variants had higher affinity against H3 HA. While variants that bound strongly to H1 HA maintained a reasonable affinity towards H3 HA, many variants that bound strongly to H3 HA exhibited a large reduction in affinity against H1 HA. For example, VVDAGW and VVEAGW

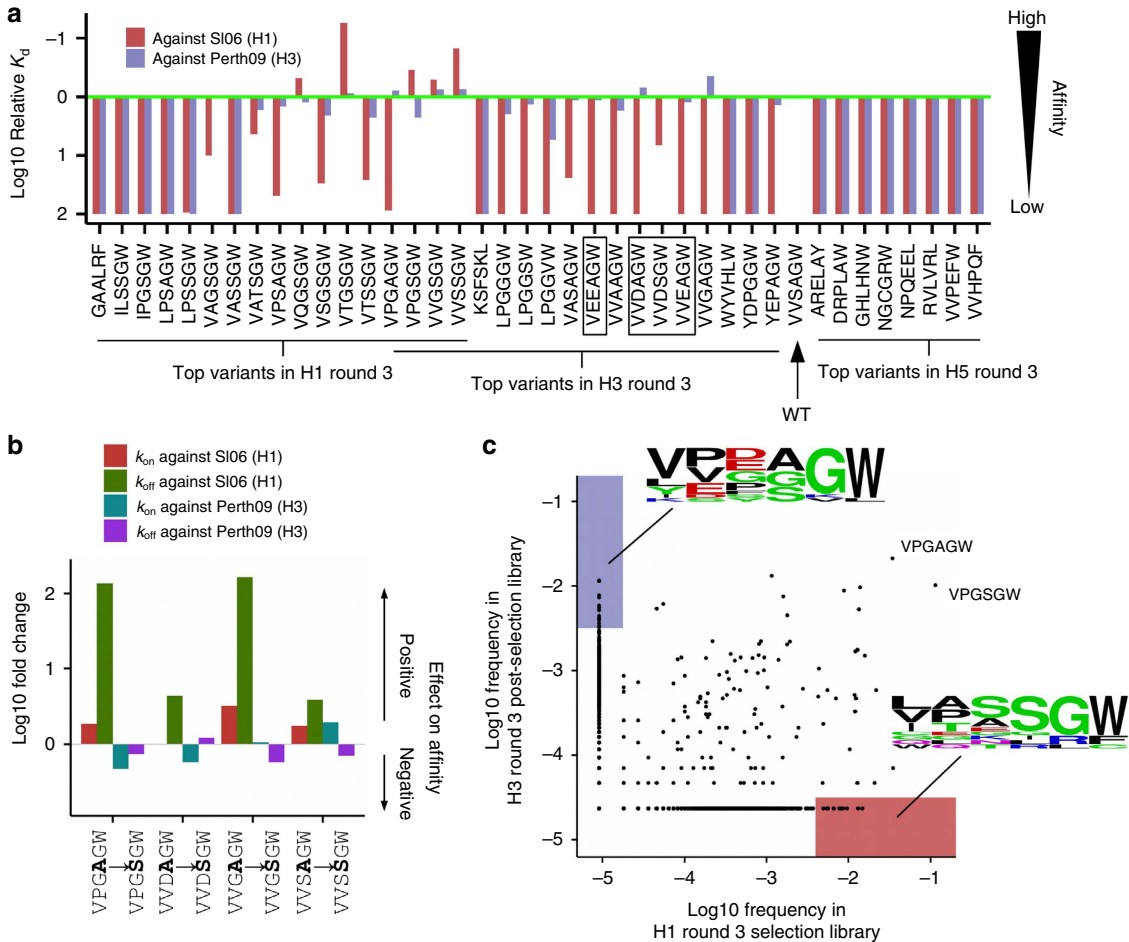

**Figure 3 | Validation for C05 variants binding against H1 HA and H3 HA.** (**a**) Supernatant of 293T cells transfected with plasmids encoding the indicated C05 Fab variants was used to estimate the affinity of C05 Fab variants against immobilized SI06 HA (H1) and A/Perth/16/2009 (Perth09) HA (H3). The relative $K_d$ of WT C05 Fab was set as 1, which is indicated by the green line. For visualizing purpose, the relative $K_d$ is capped at 100. Variants that had an H3-specific binding profile and cited in the main text are boxed. (**b**) The effect of changing position 100d from Ala to Ser on binding kinetics was investigated. Four pairs of C05 variants, in which variants within each pair were related by an Ala to Ser substitution at position 100d, were included in our validation experiment (**a**). We computed the fold change in $k_{on}$ and $k_{off}$ by comparing the binding kinetics in the variant that carried a Ser at position 4 against the variant that carried an Ala at position 4. For visualization, for a given pair of variants ($i \rightarrow j$), the fold change in $k_{on}$ was calculated as on rate of $j$ divided by on rate of $i$, whereas fold change in $k_{off}$ was calculated as off rate of $i$ divided by off rate of $j$, such that a positive value at the log scale indicates a positive effect in binding. (**c**) The occurrence frequencies of all observed variants in the sequencing data (a total of 848,630) in round 3 post-selection library and round 1 post-selection library are compared and shown as a scatterplot. Variants with >0.4% occurrence frequency in H1 round 3 post-selection library and <0.003% in H3 round 3 post-selection library were classified as 'H1 specialists' (shaded in red in the scatterplot). Variants with >0.3% occurrence frequency in H3 round 3 post-selection library and <0.001% in H1 round 3 post-selection library were classified as 'H3 specialists' (shaded in lilac in the scatterplot).

had WT-like affinity against H3 HA, but their affinities against H1 HA were >100-fold less than WT. This H3-specific binding may be partially attributed to the amino-acid identity at position 100c. Variants that had an acidic residue (Asp or Glu) at position 100c tended to exhibit this property (for example, VEEAGW, VVDAGW, VVDSGW and VVEAGW).

In fact, this subtype-dependent amino-acid preference could also be observed in variants with high affinity against H1 HA. Given the strong enrichment in the yeast display screening against H1 HA, Ser at position 100d became the focus of our analysis. From the panel of 40 candidates (Fig. 3a), four pairs of variants that differed by a single amino-acid substitution from Ala to Ser at position 100d were analysed (Fig. 3b). By comparing the binding kinetics between variants in each of these pairs, we found that this Ala to Ser substitution at position 100d dramatically improved the $k_{off}$ against H1 HA, but not against

H3 HA. To confirm this result, the WT C05 Fab, which contained an Ala at position 100d, and three C05 Fab variants with a Ser at position 100d (VPGSGW, VVSSGW and VTGSGW), which had the highest affinity against SI06 H1 HA, were expressed in insect cells, purified and their affinities against HA from a panel of human influenza strains were measured. These three variants had a much higher affinity against HAs from certain H1 strains than from H2 and H3 strains (Table 1; Supplementary Figs 5 and 6). These results further suggest that the affinity improvement from introducing a Ser at position 100d is H1-specific. Of note, the results in Fig. 3a and Table 1 were obtained from slightly different experimental systems (see Methods) and, therefore, some variation was expected.

HA subtype-specific binding could also be inferred from the next-generation sequencing data from the yeast display screening. Many variants were enriched in the selection against either H1

**Table 1 | Binding affinity of CO5 variants against hemagglutinins from different influenza strains.**

| $K_d$ in nM | Residue 190 | VVSAGW (WT) | VPGSGW | VVSSGW | VTGSGW |
|---|---|---|---|---|---|
| A/Solomon Islands/3/2006 (H1N1) | D | 91.4 ± 0.9 | 15.0 ± 0.2 | 4.6 ± 0.6 | 115.8 ± 1.9 |
| A/New Caledonia/20/1999 (H1N1) | N | 6.3 ± 0.1 | 0.6 ± 0.0 | 2.9 ± 0.1 | 1.1 ± 0.2 |
| A/Beijing/262/1995 (H1N1) | V | 23.1 ± 0.6 | 12.6 ± 0.7 | 259.3 ± 2.8 | 182.3 ± 0.5 |
| A/WSN/1933 (H1N1) | E | 169.4 ± 1.0 | 719.6 ± 2.9 | 727.7 ± 20.5 | 1,539 ± 42 |
| A/Japan/305/1957 (H2N2) | E | 1,701 ± 46 | >5,000 | >5,000 | >5,000 |
| A/Hong Kong/1/1968 (H3N2) | E | 327.8 ± 12.0 | 958.8 ± 45.6 | 1,100 ± 94 | 1,860 ± 83 |
| A/Panama/2007/1999 (H3N2) | D | 926.3 ± 12.1 | 1,306 ± 12 | 1,290 ± 71 | 4,335 ± 207 |
| A/Perth/16/2009 (H3N2)* | D | 23.4 ± 0.1 | 43.6 ± 1.0 | 43.7 ± 0.5 | 125.1 ± 1.2 |

*C05 is able to make additional interaction with A/Perth/16/2009 (H3N2) via CDR H1 (ref. 21).

HA or H3 HA, but not both (Fig. 3c). These variants are denoted as 'specialists'. Consistent with the results described above, a Ser at position 100d was enriched in H1 specialists and negatively charged residues (Asp and Glu) are enriched at position 100c in H3 specialists. Collectively, our results demonstrate that the amino-acid preference of RBS-targeted residues in C05 can differ against different HA subtypes, in which certain amino-acid substitutions in the paratope can bias the binding affinity towards a particular HA subtype.

**Structural characterization of VPGSGW in complex with HA1.** To facilitate the atomic understanding of the subtype-dependent amino-acid preference in C05, we determined the structure of HA1 subunit from A/Hong Kong/1/1968 (H3N2) in complex with a variant that carried a Ser at position 100d, namely VPGSGW, to 1.97 Å resolution (Table 2). VPGSGW was one of the most enriched variants in the yeast display screening against both H1 and H3 HAs (Fig. 3c; Supplementary Fig. 4). WT C05 and VPGSGW had a similar angle of approach and overall conformation (Fig. 4a), which is consistent with the negative-stain EM model of VPGSGW bound to the HA trimer from A/Solomon Islands/3/2006 (H1N1; Supplementary Fig. 7). Two VPGSGW-HA1 complexes were observed in the asymmetric unit. In both complexes, the electron density for the six amino-acid residues of interest was well defined (Supplementary Fig. 8). Interestingly, the side chain of S100d exhibited different conformations in the two VPGSGW-HA1 complexes (Fig. 4b), either facing outward from the six-residue loop (towards the HA RBS, conformation 1), or facing inward within the CDR H3 loop (conformation 2). Two main-chain–main-chain hydrogen bonds are present in the six-residue loop—between V100a and W100f, and between V100a and S100d. When the side chain of the Ser at position 4 faces inward, it forms two additional hydrogen bonds—one with the main-chain carbonyl of V100a and the other with the main-chain amide of W100f. These extra hydrogen bonds in conformation 2 would likely help further rigidify the six-residue loop.

Although two less intra-loop hydrogen bonds are made when S100d points towards the HA RBS (conformation 1), the Ser hydroxyl takes part in an ion–dipole network with E190 and S228 in the HA RBS (Fig. 4c). The short bond distances suggest that a sodium ion, instead of a water molecule, mediates the interaction among HA1 S228, HA1 E190, and VPGSGW S100d. Our high-resolution crystal structure also reveals that the main-chain carbonyl of G100c interacts with G225, S227 and E190 via water-mediated hydrogen bonds. In addition, similar to WT C05 (ref. 21), the main-chain carbonyls of S100d and G100e hydrogen bond with Y98 and S136, respectively. As described previously[31], it is common for HA RBS-targeted bnAbs to mimic some aspects of the binding mode of sialic acid, the natural

receptor for influenza A virus. Besides utilizing one face of the pyranose ring, sialic acid interacts with HA RBS via three moieties, namely the acetamide, carboxylate, and glycerol groups (Supplementary Fig. 9a). As seen in WT C05, VPGSGW employs the indole of W100f to bind in the same pocket as the acetamide group and the main-chain carbonyl of G100e to partially mimic the carboxylate group. However, unlike WT C05, which only partially mimics the glycerol group using the main-chain carbonyl of residue 100d, VPGSGW extends this mimicry, where the S100d hydroxyl now takes part in an ion–dipole network as described above (Supplementary Fig. 9b). Therefore, VPGSGW seemed to improve its mimicry of sialic acid compared to WT C05.

The VPGSGW region exhibits a significant local deviation in the backbone conformation from that of WT C05 around P100b, which has no contact with HA1 (Supplementary Fig. 10). In fact, V100a, P100b, G100c and S100d form a type II β-turn (Fig. 4b). It is known that type II β-turns prefer a Pro at position $i + 1$ and a Gly at position $i + 2$ (ref. 41). This suggests that the 'PG' motif stabilizes the six-residue loop to reduce the entropic cost in binding and provides an explanation for the enrichment of 'PG' motif in top variants from the screen (Figs 2b and 3c).

**Interaction between S100d in VPGSGW and residue 190 in HA.** As compared to WT C05, VPGSGW had higher affinity against certain H1 strains but lower affinity against H2 and H3 strains (Table 1), despite appearing to be a better mimic of sialic acid. We postulated that the reduction of affinity was partly attributed to Glu at residue 190. VPGSGW had higher affinities against most tested HAs from H1 strains, except for A/WSN/1933, which differed from other tested H1 strains by a D190E substitution. Similarly, the fold reduction in affinity against HA from H2 and H3 strains was larger when residue 190 was Glu rather than Asp. There also was an approximately threefold reduction in affinity against HAs from A/Japan/305/1957 (H2N2) and A/Hong Kong/1/1968 (H3N2), which have Glu at residue 190, but only an approximately twofold reduction or less against A/Panama/2007/1999 (H3N2) and A/Perth/16/2009 (H3N2), which have Asp at residue 190. On the basis of the crystal structure, the ion–dipole network that involves S100d is tightly packed and likely strongly contributes to the binding (Fig. 5a). When an Asp was modelled at residue 190, S100d may adopt a different conformation that would relieve the steric constraints encountered with Glu at position 190 (Fig. 5b).

We then measured the affinity of VPGSGW against HA from A/Hong Kong/1/1968 (HK68) that was engineered with an E190D substitution (Supplementary Fig. 11). The affinity of VPGSGW improved ~23-fold when the E190D substitution was introduced into HK68 HA (from $K_d = 959$ nM to $K_d = 42$ nM; Fig. 5c). In contrast, WT C05 only had an approximately twofold

**Table 2 | X-ray data collection and refinement statistics.**

| | |
|---|---|
| *Data collection* | |
| Beamline | SSRL 12-2 |
| Wavelength (Å) | 0.9795 |
| Space group | $P4_3$ |
| Unit cell parameters (Å) | $a = b = 88.3$, $c = 255.0$ |
| Resolution (Å) | 50-1.97 (2.04-1.97)* |
| Unique reflections | 136,341 (13,627)* |
| Redundancy | 6.3 (6.2)* |
| Completeness (%) | 99.5 (99.5)* |
| $<I/\sigma_I>$ | 15.4 (2.0)* |
| $R_{sym}$† | 0.16 (0.82)* |
| $R_{pim}$† | 0.07 (0.35)* |
| $CC_{1/2}$‡ | 0.99 (0.72)* |
| $Z_a$§ | 2 |
| | |
| *Refinement statistics* | |
| Resolution (Å) | 50-1.97 |
| Reflections (work) | 130,202 |
| Reflections (test) | 6,751 |
| $R_{cryst}(\%)^{\|\|}/R_{free}(\%)$¶ | 18.1/20.8 |
| No. of atoms | |
| Protein | |
| HA1 | 3,987 |
| Fab | 6,846 |
| Water | 1,113 |
| Ion | 1 |
| Glycan | 56 |
| Other# | 90 |
| Average *B*-value (Å²) | |
| Protein | |
| HA1 | 37.2 |
| Fab | 30.5 |
| Water | 40.0 |
| Ion | 46.3 |
| Glycan | 74.4 |
| Other# | 77.2 |
| Wilson *B*-value (Å²) | 22.7 |
| | |
| *R.m.s.d. from ideal geometry* | |
| Bond length (Å) | 0.013 |
| Bond angle (°) | 1.60 |
| | |
| *Ramachandran statistics (%)*** | |
| Favored | 96.7 |
| Outliers | 0.2 |

r.m.s.d., root mean squared deviation.
*Numbers in parentheses refer to the highest resolution shell.
†$R_{sym} = \Sigma_{hkl}\Sigma_i|I_{hkl,i} - <I_{hkl}>|/\Sigma_{hkl}\Sigma_i I_{hkl,i}$ and $R_{pim} = \Sigma_{hkl}(1/(n-1))^{1/2} \Sigma_i|I_{hkl,i} - <I_{hkl}>|/\Sigma_{hkl}\Sigma_i I_{hkl,i}$, where $I_{hkl,i}$ is the scaled intensity of the $i^{th}$ measurement of reflection h, k, l, $<I_{hkl}>$ is the average intensity for that reflection, and $n$ is the redundancy.
‡$CC_{1/2}$ = Pearson correlation coefficient between two random half datasets.
§$Z_a$ is the number of HA1-Fab complexes per crystallographic asymmetric unit.
$\|\|R_{cryst} = \Sigma_{hkl}|F_o - F_c|/\Sigma_{hkl}|F_o| \times 100$, where $F_o$ and $F_c$ are the observed and calculated structure factors, respectively.
¶$R_{free}$ was calculated as for $R_{cryst}$, but on a test set comprising 5% of the data excluded from refinement.
#Other includes non-water solvent and cryoprotectant.
**Calculated with MolProbity[63].

improvement in affinity when E190D substitution was introduced into HK68 HA (from $K_d = 328$ nM to $K_d = 146$ nM). We further performed a neutralization assay using the IgG format of WT C05 and VPGSGW against two influenza strains, namely SI06-HA/WSN and A/Aichi/2/68. SI06-HA/WSN is a recombinant H1N1 strain with Asp at residue 190 (see Methods), whereas A/Aichi/2/68 is an H3N2 strain with Glu at residue 190. When tested against SI06-HA/WSN, VPGSGW ($EC_{50} = 1.7$ μg ml$^{-1}$) is more potent than WT C05 ($EC_{50} = 5.4$ μg ml$^{-1}$). In contrast, when tested against A/Aichi/2/68, WT C05 ($EC_{50} = 15.5$ μg ml$^{-1}$) is more potent than VPGSGW ($EC_{50} = >100$ μg ml$^{-1}$). Overall, these

results suggest that the amino-acid identity at residue 190 of the HA RBS, at least partially, modulates the binding affinity of the C05 CDR H3 loop and especially in the variant VPGSGW, illustrating how a highly conserved and functionally important substitution in the RBS shifts the amino-acid preference in the paratope of a broadly neutralizing antibody.

## Discussion

Most influenza antibodies target the globular head of haemagglutinin (HA), for which the majority of its surface is readily mutable[42,43]. The HA receptor-binding site (RBS) is a highly conserved region on the HA globular head. Recently, a number of antibodies have been shown to target the HA RBS with relatively high breadth[21–25,28–31]. However, it is unclear how this breadth is acquired and sustained during the evolution of HA RBS-targeted broadly neutralizing antibodies (bnAbs). In this study, we investigated the functional sequence space of C05 (ref. 21), a prototypic HA RBS-targeted bnAb, by focusing on the six amino-acid residues on the apex of its long CDR H3 that inserts into the HA RBS and make the most intimate contacts with the RBS.

Our results reveal that certain amino-acid substitutions in CDR H3 can bias the specificity of C05 towards H1 HA or H3 HA. For example, an Ala to Ser substitution at position 100d of wild-type (WT) C05 (from VVSAGW to VVSSGW) improves the specificity towards H1 HA, whereas a Ser to Asp substitution at position 100c (from VVSAGW to VVDAGW) biases the specificity toward H3 HA. The high-resolution structure of a C05 Fab variant, VPGSGW, in complex with an H3 HA1 revealed a tightly packed, ion–dipole network involving S100d of C05 CDR H3 and E190 of HA. In the case of VPGSGW, the binding affinity towards HA that possesses an Asp at residue 190 is much higher than HA with a Glu at residue 190, likely due to more optimal steric packing. Asp predominates at residue 190 of HAs from human H1 strains, whereas Glu is prevalent in other human and all avian subtypes. E190D is a key substitution for avian-to-human receptor specificity switch in H1 strains[44]. There is also evidence showing E190D can modulate the receptor specificity in H3 strains[45] as it completely abolished viral replication in cell culture that only expresses α2,3-sialic acid[46]. In fact, perhaps somewhat surprisingly, Asp predominates at residue 190 of HAs in human H3 strains in the past two decades (Supplementary Fig. 12a), but not in avian H3 strains (Supplementary Fig. 12b). Asp and Glu only differ by a methylene group ($CH_2$), and yet impact the receptor specificity as well as the amino-acid preference of C05. Besides residue 190, substitutions at residues 225 (in H1)[47], 226 and 228 (in H2 and H3)[48,49], and a minor shift ($<1$ Å) in the width of HA RBS[49,50] also play a role in receptor specificity. This study suggests that such subtle structural variability among strains of different subtypes and receptor specificities will likely differentially influence the amino-acid preference of the paratope during antibody evolution. At the same time, RBS-targeted antibodies may also influence the evolution of HA RBS. Some residues in the HA RBS of human influenza virus continue to mutate over time, as exemplified by residue 190 (as mentioned above, Supplementary Fig. 12a) and residue 225 in H3 viruses (Supplementary Fig. 12c), which may impact the receptor-binding ability[51]. Together, these observations permit us to speculate that there is a constant evolutionary interplay between human immunity and ongoing variation in the influenza HA RBS.

Many influenza bnAbs, including C05, have been isolated from the immune repertoires of people vaccinated or naturally infected with influenza virus[5,8,10,21,25,28,30,52]. The breadth of these bnAbs may change when undergo further affinity maturation. This study

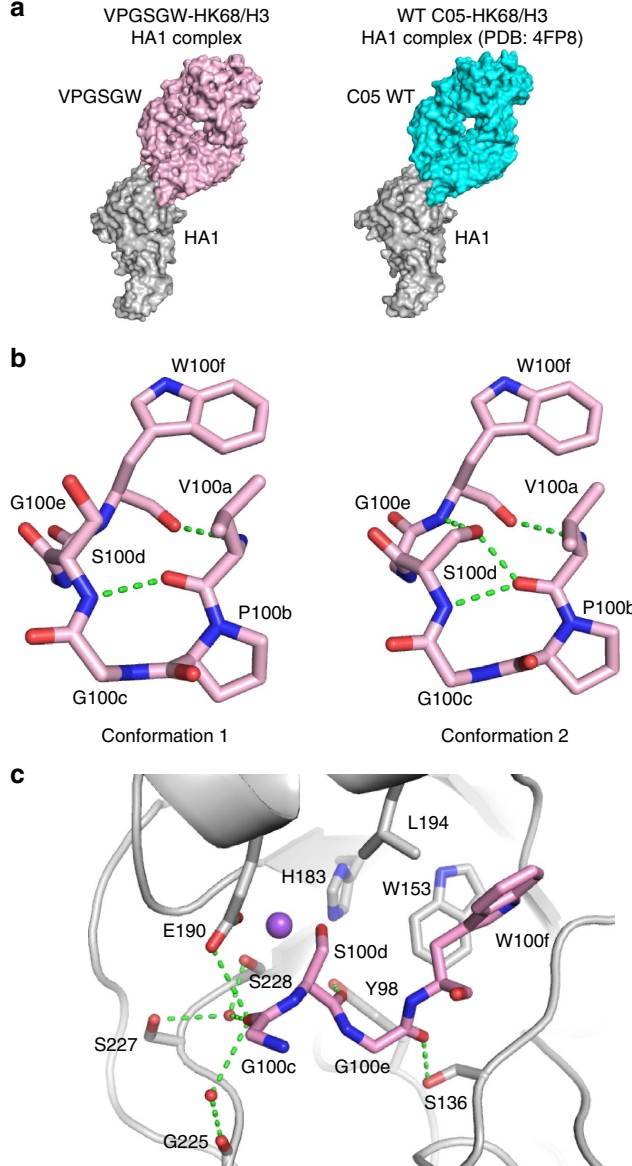

**Figure 4 | Interaction between VPGSGW and HK68/H3 HA1.** (**a**) The crystal structure of VPGSGW in complex with HA1 exhibits a similar conformation to that of C05 WT in complex with HA1 (PDB: 4FP8)[21]. (**b**) There are two VPGSGW-HA1 complexes in the asymmetric unit of the crystal structure. The orientations of the serine at 100d differ between those two complexes. Intramolecular hydrogen bonds are shown for the two different conformations of CDR H3 of VPGSGW. (**c**) Interaction of the HA RBS (grey) with CDR H3 (pink) is shown. Hydrogen bonds are represented by green dashed lines. Water molecules are represented by red spheres. A putative sodium ion is represented by the purple sphere. For clarity, only G100c, S100d, G100e, and W100f on the CDR H3 of VPGSGW are displayed.

suggests that, at least in the case of HA RBS-targeted bnAbs, a tradeoff exists between affinity and breadth. As bnAbs continue to evolve during the natural process of affinity maturation, they may gain affinity by exploiting subtle structural features that are unique to the specific subtype that originally stimulated them. Group-specific or subtype-specific structural features are also known to present in the epitopes of stem-binding bnAbs[6,8,9,11,12]. It is possible that such subtype-dependent amino-acid preferences can be observed in other HA RBS-targeted and in stem-binding

bnAbs. The breadths of these bnAbs may be unsustainable during repeat challenge of the same subtype, as they would become 'specialized'. However, this tradeoff between affinity and breadth is unlikely to be universal to all bnAbs, as the affinity maturation of certain stem-binding bnAbs is characterized by both increased breadth and potency[8,11,14].

While our results provide valuable insight into the interaction between a peptide scaffold and HA RBS, complete optimization of C05 affinity and breadth would likely also involve to some extent CDR H1 and the rest of CDR H3, which make additional contacts with neighboring regions of HA RBS that are less evolutionarily conserved[21], as well as CDR H2, which may be important for stabilizing CDR H3 (ref. 21). Nonetheless, the rest of the C05 epitope that is not examined in this study has higher sequence diversity in naturally circulating strains (Supplementary Fig. 1b), which may cause a severe tradeoff between affinity and breadth. The influence of these factors on C05 binding affinity and breadth deserves further investigation.

As compared to influenza virus, the evolution of bnAbs is better characterized in the human immunodeficiency virus (HIV) field because it is more clinically practical to follow a chronically infected patient over a long period of time. It is known HIV bnAbs usually emerge only after a few years of infection with a large amount of somatic mutations[53,54]. In comparison, the number of somatic mutations in influenza bnAbs is much fewer[55]. It is likely that our immune system have optimized the germline sequences to respond to pathogens that circulate in human for centuries[56]. Compared to influenza virus, HIV is a relatively new virus to humans, which may explain the difference in the number of somatic mutations required for developing bnAbs against these two viruses. This also highlights the role of human-virus interaction history in the evolution of bnAbs[56]. In addition, the mode of infection may play a role in how bnAbs evolve. While HIV infection is chronic, influenza infection is not. HIV bnAbs co-evolve with antigen within the same host, whereas influenza viruses co-evolve with host immunity at a population level. The distribution of antigen on the virus surface is another critical factor to be considered. It is evidenced that HA RBS-targeted bnAbs often have much lower Fab affinity than would be expected for an effective neutralizing antibody, but can make up for this lower affinity via the bivalent avidity in the IgG to achieve high neutralization breadth[23]. Such an avidity effect is attributed to the relatively close proximity of neighboring HAs on influenza virus. In contrast, envelope proteins on HIV are much more sparse, which allows HIV to escape from the antibody avidity effect[57]. Comparison between HIV bnAbs and influenza bnAbs highlights how the intrinsic properties of the virus may influence the evolution of bnAbs.

While this study is only based on only one broadly neutralizing antibody, it is likely that the tradeoff between affinity and breadth could vary among different antibody scaffolds and epitopes. For an ideal bnAb, affinity and breadth should be correlated such that affinity improvement against one subtype would be translated to other subtypes. This evolutionary aspect of bnAbs represents a critical factor to be considered for immunogen design in universal flu vaccine development. It is important not only to induce bnAbs, but also to induce bnAbs with minimal tradeoff between affinity and breadth. As there are active ongoing developments towards a universal influenza vaccine[18,19,20,58], the continued evolution of influenza bnAbs is certainly worth investigating in greater depth and detail.

## Methods

**Purification and biotinylation of influenza hemagglutinin.** Influenza HA was prepared for binding studies as previously described[7]. Briefly, the HA ectodomain was fused with an N-terminal gp67 signal peptide and a C-terminal BirA

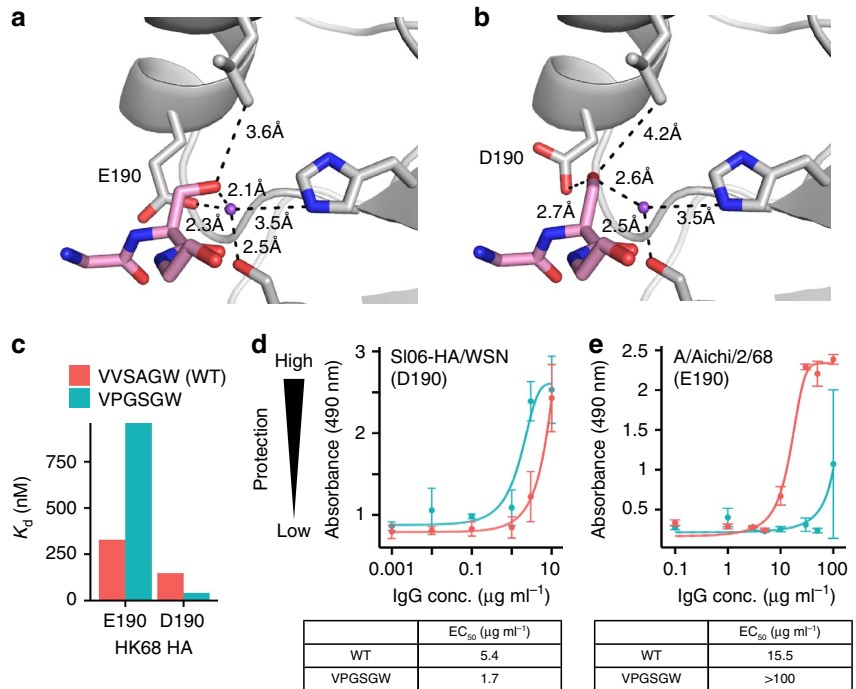

**Figure 5 | E190D in the HA RBS favors binding against VPGSGW.** The hydrogen bond network involving S100d of the CDR H3 of VPGSGW is shown for (**a**) binding against HA RBS of HK68/H3, and (**b**) binding against HA RBS of HK68/H3 that carried mutation E190D, which is modelled based on the crystal structure of VPGSGW-HK68/H3 HA1. A putative sodium ion is represented by the purple sphere. For visual clarity, only G100c, S100d, G100e, and W100f on the Fab are displayed. (**c**) The affinities of WT C05 Fab (VVSAGW) and a C05 Fab variant (VPGSGW) against the HA from A/Hong Kong/1/1968 (wild type: E190; E190D mutant: D190) are shown as a bar chart. (**d,e**) The neutralizing activity of WT C05 and VPGSGW in IgG format against (**d**) SI06-HA/WSN virus, and (**e**) A/Aichi/2/68 virus were measured by cell viability assay. SI06-HA/WSN virus was generated based on WSN, in which the HA ectodomain was replaced by that from SI06 (see Methods). Colour code is the same as that of **c**. Mean value across three replicates is shown and the error bar represents the s.d. The large error bar in VPGSGW at 100 µg ml$^{-1}$ against A/Aichi/2/68 virus is due to complete protection in one but not in the other two replicates.

biotinylation site, thrombin cleavage site, trimerization domain, and a His$_6$ tag and cloned into a customized baculovirus transfer vector[7]. Recombinant bacmid DNA was generated using the Bac-to-Bac system (Life Technologies, Carlsbad, CA). Baculovirus was generated by transfecting purified bacmid DNA into Sf9 cells using FuGene HD (Promega, Madison, WI). HA was expressed by infecting suspension cultures of High Five cells (Life Technologies) with baculovirus at an MOI of 5 to 10 and incubating at 28 °C shaking at 110 r.p.m. for 72 h. The supernatant was concentrated. HA0 was purified by Ni-NTA and buffer exchanged into 20 mM Tris-HCl pH 8.0 and 150 mM NaCl. Biotinylation was performed by incubating 25 µg of BirA enzyme per 1 mg of HA in 100 mM Tris pH 8.0, 10 mM ATP, 10 mM MgOAc, 50 µM biotin and 50 mM NaCl as described previously[21]. Biotinylated HAs were purified by size exclusion chromatography.

**Construction of C05 Fab mutant library.** The insect cell expression plasmid that encodes WT C05 Fab[21] was used as the template to generate the C05 Fab mutant library. PCR was performed using KOD DNA polymerase (EMD Millipore, Billerica, MA) with 1.5 mM MgSO$_4$, 0.2 mM of each dNTP (dATP, dCTP, dGTP and dTTP), and 0.5 µM each of the primers 5′-AAGCACATGTCAATGCAG CAANNKNNKNNKNNKNNKTDKGAAAGAGCCGATTTGGTGGGT-3′ (C05HC-lib-F) and 5′-AGCGTAGTCTGGAACGTCGTATGGGTACAGGC CCCCGAGGCCACAGGATTTTGGCTCGACTCTTTTGTCGAC-3′ (C05HC-NTD1-R) were used. Another PCR was performed using the same condition except that primers 5′-GCTTCAGTTTTAGCAGCGGCCCAGCCGGCCCAGGTCCA ACTCCAAGAAAGCGGC-3′ (C05HC-NTD1-F) and 5′-TTGCTGCAT TGACATGTGCTTAGCGC-3′ (C05HC-Frag1-R) were used instead. The products from these two PCRs were purified by PureLink PCR Purification Kit (Life Technologies) according to manufacturer's instructions. An overlapping PCR was performed using 10 ng each of the purified product. The condition for the overlapping PCR was the same as above, except that C05HC-NTD1-F and C05HC-NTD1-R were used as primers. The product from the overlapping PCR was purified by PureLink PCR Purification Kit (Life Technologies) according to manufacturer's instructions and digested with SfiI (New England Biolabs, Ipswich, MA) for 10 h at 50 °C. The digested product was purified by gel extraction using PCR Clean-Up and Gel Extraction Kit (Clontech Laboratories, Mountain View, CA) according to manufacturer's instructions. Ligation to the SfiI-digested dual promoter yeast expression vector was performed using T4 DNA ligase

(New England Biolabs). The ligated product was transformed into MegaX DH10B T1R cells (Life Technologies). Around 10$^8$ colonies were collected. C05 Fab plasmid mutant library were purified from the bacteria colonies using Maxiprep Plasmid Purification (Clontech Laboratories).

**Yeast display screening and FACS.** The coding sequence of the heavy chain of the C05 Fab plasmid mutant library was amplified by PCR using KOD DNA polymerase (EMD Millipore) with 1.5 mM MgSO$_4$, 0.2 mM of each dNTP (dATP, dCTP, dGTP and dTTP), and 0.5 µM each of the primers 5′-GCTTCAG TTTTAGCAGCGGCCCAG-3′ and 5′-AGCGTAGTCTGGAACGTCGTATGG GTACAGGC-3′. The PCR product (C05 library heavy-chain insert) was purified by gel extraction using PCR Clean-Up and Gel Extraction Kit (Clontech Laboratories) according to the manufacturer's instructions. Yeast transformation was performed by following the LiAc/SS-DNA/PEG protocol described in (http://mcb.berkeley.edu/labs/koshland/Protocols/YEAST/LiAc.html). Four micrograms of the SfiI-digested dual promoter yeast expression vector and 8 µg of the C05 library heavy-chain insert were used for the transformation. Transformants were incubated at 30 °C for 2 days. At least 10$^8$ colonies were collected, resuspended in YPD with 15% glycerol, and stored at − 80 °C until used.

For each round of enrichment, ∼10$^9$ yeast cells from the frozen stock were cultured in 250 ml SDCAA (2.0% glucose, 0.67% yeast nitrogen base, 0.5% casamino acids, 0.54% disodium phosphate and 0.86% monosodium phosphate) for 18 h at 28 °C with shaking at 250 r.p.m. The initial OD$_{600}$ was ∼0.3 and the final OD$_{600}$ was ∼1.7. Yeast cells were spun down at 4 °C with 4,750 r.p.m. for 20 min and resuspended in 100 ml SGR-CAA (20 g l$^{-1}$ galactose, 20 g l$^{-1}$ raffinose, 1 g l$^{-1}$ dextrose, 6.7 g l$^{-1}$ yeast nitrogen base, 5 g l$^{-1}$ casamino acids, 5.4 g l$^{-1}$ Na$_2$HPO$_4$ and 8.56 g l$^{-1}$ NaH$_2$PO$_4$) with an OD$_{600}$ of ∼0.6. Yeast cells were cultured for 24 h at 18 °C with shaking at 250 r.p.m. to reach an OD$_{600}$ of 1.5. 15 ml of the yeast culture was spun down, washed twice with PBS, and resuspended in 5 ml PBS. Biotinylated trimeric HA was incubated with Streptavidin PE (eBioscience, San Diego, CA) at a molar ratio of 1:4 for 15 min. Of note, Streptavidin PE was buffer exchanged into PBS before use to remove NaN$_3$, which is toxic to the yeast cells. The biotinylated trimeric HA-strepavidin PE complex was added to the yeast cells in PBS with a final concentration as indicated in Supplementary Table 1. After incubating at 4 °C overnight with head-to-head rotation, the yeast cells were spun down, washed twice with PBS, resuspended in

5 ml PBS, and subjected to fluorescence-activated cell sorting at TSRI Flow Cytometry Core Facility. The sorted yeast cells were recovered by plating on the SDCAA agar plates. Yeast colonies were collected after 2 days of incubation at 30 °C, resuspended in YPD with 15% glycerol, and stored at $-80$ °C until use.

**Sequencing library preparation.** Plasmid was extracted from at least $10^7$ yeast cells per sample using Zymoprep Yeast Plasmid Miniprep II (Zymo Research, Irvine, CA) according to the manufacturer's instructions. The mutated region on the C05 heavy chain was amplified by PCR using KOD DNA polymerase (EMD Millipore) with 1.5 mM $MgSO_4$, 0.2 mM of each dNTP (dATP, dCTP, dGTP and dTTP), and 0.5 μM each of the primers 5′-CACTCTTTCCCTACACGA CGCTCTTCCGATCTACCGGTGTGTACTACTGCGCTAAGC-3′ and 5′-GACT GGAGTTCAGACGTGTGCTCTTCCGATCTTCCCCACACATCGAAGGCG TCACCC-3′. The product was purified with a PureLink PCR Purification Kit (Life Technologies, Carlsbad, CA) according to manufacturer's instructions. The purified product was used as a template for a second PCR performed under the same conditions except that primers 5′-AATGATACGGCGACCACCGAGATC TACACTCTTTCCCTACACGACGCT-3′ and 5′-CAAGCAGAAGACGGCATA CGAGAT<u>NNNNNNN</u>GTGACTGGAGTTCAGACGTGTGCT-3′ were used instead. The underlined 'N's indicate the position of the barcode for multiplex sequencing. The sequences of the barcodes are indicated in Supplementary Table 2. The product from the second PCR was subjected to next-generation sequencing using Illumina MiSeq $2 \times 75$ bp paired-end reads at the TSRI Next Generation Sequencing Core.

**Sequencing data analysis.** The sequencing data were demultiplexed using the barcode reads. For each paired-end read, the nucleotide sequence corresponding to the randomized region was extracted. If the nucleotide sequence at the randomized region was inconsistent between forward and reverse reads, the paired-end read would be discarded. In other words, at the randomized region, the reverse-complement of forward read must perfectly match the reverse read. The nucleotide sequence was translated into the amino-acid sequence. The occurrence of each amino-acid sequence was counted, with each paired-end read as one count. WT C05 Fab from incomplete digestion of the vector during mutant library construction can be distinguished from the WT C05 Fab in the mutant library due to differences in codon usage. Those counts corresponding to WT C05 Fab from incomplete digestion of the vector were filtered from downstream analysis unless otherwise stated. Custom python scripts were used for sequencing data processing. All scripts have been deposited to https://github.com/wchnicholas/C05mut.

**Construction and purification of C05 Fab variants.** Individual mutants for the validation experiment were constructed using the QuikChange XL Mutagenesis kit (Stratagene, San Diego, CA) according to the manufacturer's instructions. Primers for Quikchange were designed such that they matched 18 bp flanking each side of the mutated region. The nucleotide sequence of the mutated region on the primers was designed to minimize nucleotide mismatch with the WT C05 Fab. For the expression of C05 Fab variants in mammalian cells, both light and heavy chains were cloned into the pFuse vector, which was constructed by removing the Fc-encoding sequence from the pFuse-Fc vector (Invivogen, San Diego, CA). The light chain and heavy chain were transfected into 293T cells in a 2:1 molar ratio. A $His_6$-tag was inserted at the C-terminus of the heavy chain. The supernatants containing the C05 Fab variants were collected 3 days after transfection. Expression of C05 Fab variants in insect cells was performed as described previously for WT C05 Fab[21]. Purification of C05 Fab variants was performed by Ni-NTA Superflow (Qiagen, Valencia, CA), and subsequently by size exclusion chromatography on a Hiload 16/90 Superdex 200 column (GE Healthcare, Pittsburgh, PA) in 20 mM Tris pH 8.0, 150 mM NaCl, and 0.02% $NaN_3$.

**Biolayer interferometry binding assay.** The binding assay was performed by biolayer interferometry (BLI) using an Octet Red instrument (ForteBio, Menlo Park, CA). Biotinylated HA0 at approximately 10–50 μg ml$^{-1}$ in $1 \times$ kinetics buffer ($1 \times$ PBS with 0.01% BSA and 0.002% Tween 20) was loaded onto streptavidin biosensors and incubated with supernatant from transfected cells or with the indicated concentration of Fab. Streptavidin biosensors that were not loaded were used as a reference for subtracting background binding from signals. Briefly, the assay consisted of five steps: (1) baseline: 60 s with $1 \times$ kinetics buffer; (2) loading: 120 s with biotinylated HA0; (3) baseline: 60 s with $1 \times$ kinetics buffer; (4) association: 120 s with samples (supernatant from transfected cells or purified Fab); and (5) dissociation: 120 s with $1 \times$ kinetics buffer. For binding assay that used supernatant from transfected cells, relative Fab concentrations were determined by western blot using a monoclonal antibody to the His-tag (catalogue number: MAB230P, Maine Biotechnology, Portland, ME) as the primary antibody, anti-mouse goat antibody (catalogue number: 115-035-008, Jackson ImmunoResearch, West Grove, PA) as the secondary antibody, and subsequent densitometry analysis using ImageJ. Since the concentration of C05 Fab in the supernatant was unknown, we were not able to calculate the exact $K_d$. Instead, we normalized the apparent $K_d$ to that of WT C05 Fab (VVSAGW) and further normalized to the expression level to estimate the relative $K_d$. The relative $K_d$ of WT C05 Fab was set as 1. For estimating the exact $K_d$, a 1:1 binding model was used. In cases where the

binding affinity was relatively weak ($K_d > 300$ nM), a 1:1 binding model did not fit well due to the contribution of non-specific binding to the response curve. Subsequently, a 2:1 heterogeneous ligand model was used to improve the fitting.

**Crystallization and structural determination.** HA1 (H3 numbering: residues 43–309) from A/Hong Kong/1/1968 (HK68/H3) was expressed in insect cells as described[21] and purified by Ni-NTA Superflow (Qiagen) and subsequently by size exclusion chromatography on a Hiload 16/90 Superdex 200 column (GE Healthcare) in 20 mM Tris pH 8.0, 150 mM NaCl, and 0.02% $NaN_3$. The C05 variant VPGSGW was incubated with HK68/H3 HA1 in a molar ratio of 1.5:1 overnight at 4 °C. The VPGSGW-HK68/H3 HA1 complex was purified by size exclusion chromatography on a Hiload 16/90 Superdex 200 column (GE Healthcare) in 20 mM Tris pH 8.0, 150 mM NaCl, and 0.02% $NaN_3$ and concentrated to $\sim 10$ mg ml$^{-1}$ in 10 mM Tris pH 8.0, 50 mM NaCl, and 0.02% $NaN_3$. Crystal screening was carried out using our high-throughput, robotic CrystalMation system (Rigaku, Carlsbad, CA) at TSRI. Further optimization was based on the initial hit using the sitting drop vapour diffusion method with 500 μl reservoir solution containing 0.1 M sodium citrate pH 5.5 and 9% PEG 8000. Drops consisting 0.8 μl protein + 0.8 μl precipitant were set up at 20 °C and crystals appeared within a week. The resulting crystals were cryoprotected by soaking in well solution supplemented with 15% PEG 400, flash cooled, and stored in liquid nitrogen until data collection.

Diffraction data for the VPGSGW-HK68/H3 HA1 complex were collected at the Stanford Synchrotron Radiation Lightsource beamline 12-2. The data were indexed in space group $P4_3$, and integrated and scaled using HKL2000 (HKL Research, Charlottesville, VA)[59]. The structure was solved by molecular replacement at 1.97 Å resolution using Phaser[60] with PDB 4FP8 (ref. 21) as the molecular replacement model, modelled using Coot[61], and refined using Refmac5 (ref. 62). Ramachandran statistics were calculated using MolProbity[63].

**Electron microscopy reconstruction.** HA0 from A/Solomon Islands/3/2006 (SI06/H1) was treated with trypsin (New England Biolabs) to remove the C-terminal tags and cleave to produce mature HA. The trypsin-digested HA was then purified by size exclusion chromatography. VPGSGW was incubated with the purified HA in a molar ratio of 4.5:1 overnight. The VPGSGW-HA complex was purified by size exclusion chromatography. CR9114 Fab[8] was then incubated with the purified VPGSGW-HA complex in a molar ratio of 4.5:1. Of note, additional VPGSGW was not supplied for this incubation and may explain the low occupancy of VPGSGW in the negative-stain data (Supplementary Fig. 7). The CR9114-VPGSGW-HA complex was purified by size exclusion chromatography. All size exclusion chromatography were performed on a Hiload 16/90 Superdex 200 column (GE Healthcare) in 20 mM Tris pH 8.0, 150 mM NaCl, and 0.02% $NaN_3$. The purified CR9114-VPGSGW-HA complex was added onto 400 mesh carbon coated copper grids with 2% uranyl formate. The grid was imaged on a Tecnai Spirit with camera at $\sim 1.5$ μm defocus. The micrographs were collected using the program Leginon[64], processed using Appion[65], and particles were selected using DogPicker[66]. Particles were aligned into two-dimensional (2D) classes and EMAN2 (ref. 67) was used to generate an initial model. Using the model along with a clean stack of particles, which contained 18,874 particles at 2.05 Å per pixel, a final 3D reconstruction was produced at resolution of 11 Å (FSC = 0.5). EMD maps or PDB models were docked into the density using Chimera[68].

**IgG expression and purification and neutralization assay.** The heavy chains and light chains of C05 (WT or VPGSGW) were cloned into pFuse vector separately, under control of EF1a promoter and HTLV enhancer. The plasmids were co-transfected into Expi293F cells at a 2:1 ratio (light: heavy), The supernatant was collected at 72 h post transfection. Full-length IgG proteins were purified from the supernatant using protein G column on AKTAexpress (GE Healthcare).

SI06-HA/WSN virus was rescued using the A/WSN/33 eight-plasmid reverse genetics system[69], with the HA ectodomain of WSN (H3 numbering: HA1 residue 13 to HA2 residue 171) replaced with that of SI06. A/Aichi/2/68 (H3N2) virus was a kind gift from Andrew Thompson and James Paulson at The Scripps Research Institute.

Around 10,000 WT or mutant viruses were incubated with IgG at the indicated concentration for 1 h. The virus-IgG mixture was added to MDCK-SIAT1 (for SI06-HA/WSN) or MDCK cells (for A/Aichi/2/68) in a 96-well plate. Cells were seeded at a density of 5,000 cells per well 1 day prior to infection. At 3-day post-infection, cell viability was measured using the CellTiter-Glo Luminescent Cell Viability Assay (Promega) according to manufacturer's instructions.

**Generation of sequence logo.** Sequence logos were generated by WebLogo (http://weblogo.berkeley.edu/logo.cgi)[70].

**Data availability.** Raw sequencing data have been submitted to the NIH Short Read Archive under accession number: BioProject PRJNA326694. The x-ray coordinates and structure factors have been deposited in the RCSB Protein Data Bank under accession code 5UMN. The EM map has been deposited in the EMDB

under accession number EMD-8578. All the other data that support the conclusions of the study are available from the corresponding author upon request.

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

## Acknowledgements

We thank Kristen Slanina, Matthew Haynes, and Brian Seegers of the TSRI Flow Cytometry Core Facility for performing FACS, Steven Head, Jessica Ledesma and Lana Schaffer at TSRI Next Generation Sequencing Core for next-generation sequencing, Xueyong Zhu and Steffen Bernard for assistance with X-ray crystallography data processing, Shanshan Lang for providing CR9114 Fab, and Andrew Thompson and James Paulson for providing A/Aichi/2/68 (H3N2) virus. We acknowledge NIH R56 AI117675 for support. N.C.W. was supported by the Croucher Foundation Fellowship.

## Author contributions

N.C.W., J.X., A.B.W., R.A.L. and I.A.W. conceived and designed the experiments, N.C.W., G.G. and J.X. designed and performed the yeast display screening and validation experiments, N.C.W. performed the X-ray data collection, structure determination and refinement, H.L.T. and A.B.W. performed the EM experiments, N.C.W., J.X., A.B.W., R.A.L. and I.A.W. analysed the data and wrote the manuscript. All authors edited the paper.

## Additional information

**Competing interests:** The authors declare no competing financial interests.

**Publisher's note**: 

