## [Peer Review File · Nature Communications]

Reviewers' comments:

Reviewer #1 (Remarks to the Author):

This manuscript presented the results of a rather extensive set of experiments, including yeast display screening, investigating a region of six residues in the HCDR3 of C05, a bnAb against influenza that has been shown to bind the HA receptor binding pocket predominantly by its long HCDR3. One might hope that such experiments will lead to improved breadth and potency of the original bnAb; it's rather disappointing that they only generated subtype specific variants with high potencies. There are several points the authors should address to improve the manuscript. 1) Why the region of only the six apex residues HCDR3 was selected? There are certainly additional residues of the HCDR3 involving in antibody binding. Is the selection limited by the methodology or by design? 2) Although the original structural manuscript of C05 included sequence conservation of the receptor binding site, it'll be useful to show a sequence logo of the contact residues in the pocket, in particular those involved in binding the six residues. 3) Would such sequence analysis provide any clue that there is ever any chance to improve the breadth of the bnAb? Would any of the variants (not tested) that might have improved breadth? 4) How much might the yeast display mimic the SHM of ab maturation?

Reviewer #2 (Remarks to the Author):

Summary:

In this study We et al. investigated the functional sequence space of C05, a very well characterized anti-globular head antibody binding to the receptor binding site, by focusing on the six amino-acid residues on the apex of the C05 long HCDR3 that inserts into the HA RBS and is the key element of C05-mediated neutralization.

Saturated mutagenesis of the paratope region (6 amino-acids) of C05 was used to create a yeast display library of C05 mutants that was selected by parallel screenings using either H1 or H3 HAs. Using this approach Authors identified C05 variants with increased affinity for the selecting antigens used (i.e. H1 or H3). The analysis of the selected C05 variants showed that specific amino-acid substitutions in CDR H3 can influence the specificity of C05 towards H1 HA or H3 HA. Of note, some variants showed improved affinity of more than 20-fold. However, this increased affinity was accompanied by a significant loss of breadth.

Through crystal structure analysis of C05 Fab variant with increased affinity in complex with the H1 subunit of the HA trimer, they demonstrated that the HA subtype preference of the H1 increased affinity C05 variant is, at least in part, attributed to the amino-acid identity at residue 190 in the HA RBS. Indeed, although a large portion of the HA RBS is extremely conserved across subtypes, some minor, but important, sequence and structural variations exist in the RBD and these can affect receptor-binding preferences among different subtypes.

The authors concluded that, at least for HA RBS-targeted bnAbs, a tradeoff between breadth and affinity might exist.

In general, the manuscript is well written and a considerable amount of experimental data

are provided to support the hypothesis that subtle variations in the HA RBS between subtypes can differentially influence the evolution of high affinity RBS BNABs.

Major points:

- Line 54-55: it would be important mentioning that anti-stem antibodies have not just been used to guide vaccine development but that some are currently in clinical development as antivirals.
- Lines 115-116: it would be worth mentioning that the breadth is characterized by the neutralization of several (not all) strains tested within each subtype. In addition, it would be worth mentioning that only a few (and amongst those being C05 the most important example) of the RBS antibodies are endowed with heterosubtypic activity (i.e. they are broad but within a specific subtype).
 - A general criticism arises from the lack of analysis on the role of somatic mutations outside C05 HCDR3. In particular, in Ekiert et al. Nature 2012 it was shown that C05 carries a five-AA insertion in HCDR1 and that, in spite of making only minor contacts with HA, this insertion plays a role in the recognition with high affinity of contemporary H3 isolates. In addition, the highly mutated HCDR2 was shown to have important contact sites with the backside of HCDR3, with a possible stabilizing role. It would be important to evaluate the role of somatic mutations events (including the 5-AA insertion) in combination with the in vitro optimized HCDR3 variants for instance by testing these new variants in combination with the germlined backbone or with C05 variants selectively germlined in HCDR1 or HCDR2. This analysis would allow to put in a more biological context the analysis of C05 evolution, that is here just limited to an in vitro affinity maturation exercise. The understanding of the importance of the somatic insertion in HCDR1 is important for instance for a better understanding of the key evolution events required for the development of universal influenza vaccines targeting RBD.
 - In the discussion (lines 363-365) it is suggested that antibody evolution and increased affinity is combined with a progressively reduced breadth. While the Authors are not tempted to generalize, it might be important mentioning other examples related to anti-stem antibodies, whereby affinity maturation is characterized by both increased breadth and potency (e.g. group 1 anti-stem antibodies evolving into group 1+2).

Minor points:

- It is not always clear if the 3 round of panning have been made with the 3 viruses (H1, H3, H5). In Fig. 1 for example this detail should be mentioned, likewise in line 146.
- Figure 5 legend: in the description of (e) panel the name of the corresponding virus is missing (i.e. A/Aichi/2/68)
- Suppl. Fig. 9: the legend should indicate that sialic acid is depicted in yellow

Reviewer #3 (Remarks to the Author):

Wilson and coworkers investigated a broadly neutralizing antibody that is directed against the receptor binding site of influenza hemagglutinin. They randomized the tip of the CDR3 paratope, generated a yeast display library and screened for binders with enhanced affinity to H1 HA or H3 HA, respectively. Statistical analysis of next generation sequencing data after screening round 3 revealed various types of variants. The authors generated a cocrystal of H1 HA with an affinity improved variant and thereby gained insight into subtype specificity. The data are technically sound and the combination of YSD screening and NGS provided a wealth of information on sequence preference with respect to subtype binding. While the obtained variants are of no direct value for antiinfective therapy, they provide valuable insight into evolutionary drift of subtype HA RBS. I can recommend the paper for publication without changes.

Reviewers' comments:

Reviewer #1 (Remarks to the Author):

This manuscript presented the results of a rather extensive set of experiments, including yeast display screening, investigating a region of six residues in the HCDR3 of C05, a bnAb against influenza that has been shown to bind the HA receptor binding pocket predominantly by its long HCDR3. One might hope that such experiments will lead to improved breadth and potency of the original bnAb; it's rather disappointing that they only generated subtype specific variants with high potencies. There are several points the authors should address to improve the manuscript.

1) Why the region of only the six apex residues HCDR3 was selected? There are certainly additional residues of the HCDR3 involving in antibody binding. Is the selection limited by the methodology or by design?

Response: Thank you for the comment. Indeed, the experimental design is limited to some extent by the screening methodology. Yeast display screening has an estimated maximum throughput limit of ~100 million. Therefore, we only focused on the six apex that are the main contact and buried residues in the HA receptor binding site that resulted in a total mutant library diversity of ~20 million, which is within the throughput limit. This is stated in the revised manuscript in line 140-142: "The resultant mutant library has an amino-acid sequence diversity of ~20 million variants ($20^5 \times 6 = 19,200,000$), within the estimated throughput limit (~100 million) for yeast display screening³⁹."

2) Although the original structural manuscript of C05 included sequence conservation of the receptor binding site, it'll be useful to show a sequence logo of the contact residues in the pocket, in particular those involved in binding the six residues.

Response: Thank you for the suggestion. In the revised manuscript, we have included a sequence logo of the residues in the C05 epitope as Supplementary Figure 1b.

3) Would such sequence analysis provide any clue that there is ever any chance to improve the breadth of the bnAb? Would any of the variants (not tested) that might have improved breadth?

Response: The rest of the C05 epitope has high sequence diversity. Therefore, it may pose a huge challenge in simultaneously optimizing the breadth and affinity of C05 by mutations outside of the six residues of interest. This is discussed in the revised manuscript in line 378-380: "Nonetheless, the rest of the C05 epitope that is not examined in this study has higher sequence diversity in naturally circulating strains (Supplementary Fig. 1b), which may cause a severe tradeoff between affinity and breadth."

4) How much might the yeast display mimic the SHM of ab maturation?

Response: The mutant library in our yeast display screening experiment should explore a larger sequence diversity than that generated by SHM. This is discussed in the revised manuscript in line 142-144: "The sequence diversity of these six residues of interest in the mutant library should be higher than that generated by somatic hypermutation, which

is mostly limited to substitutions resulting from single nucleotide changes.”

Reviewer #2 (Remarks to the Author):

Summary:

In this study We et al. investigated the functional sequence space of C05, a very well characterized anti-globular head antibody binding to the receptor binding site, by focusing on the six amino-acid residues on the apex of the C05 long HCDR3 that inserts into the HA RBS and is the key element of C05-mediated neutralization. Saturated mutagenesis of the paratope region (6 amino-acids) of C05 was used to create a yeast display library of C05 mutants that was selected by parallel screenings using either H1 or H3 HAs. Using this approach Authors identified C05 variants with increased affinity for the selecting antigens used (i.e. H1 or H3). The analysis of the selected C05 variants showed that specific amino-acid substitutions in CDR H3 can influence the specificity of C05 towards H1 HA or H3 HA. Of note, some variants showed improved affinity of more than 20-fold. However, this increased affinity was accompanied by a significant loss of breadth. Through crystal structure analysis of C05 Fab variant with increased affinity in complex with the H1 subunit of the HA trimer, they demonstrated that the HA subtype preference of the H1 increased affinity C05 variant is, at least in part, attributed to the amino-acid identity at residue 190 in the HA RBS. Indeed, although a large portion of the HA RBS is extremely conserved across subtypes, some minor, but important, sequence and structural variations exist in the RBD and these can affect receptor-binding preferences among different subtypes. The authors concluded that, at least for HA RBS-targeted bnAbs, a tradeoff between breadth and affinity might exist.

In general, the manuscript is well written and a considerable amount of experimental data are provided to support the hypothesis that subtle variations in the HA RBS between subtypes can differentially influence the evolution of high affinity RBS BNABs.

Response: Thanks for the supportive comments.

Major points:

- Line 54-55: it would be important mentioning that anti-stem antibodies have not just been used to guide vaccine development but that some are currently in clinical development as antivirals.

Response: We agree that this is an important point, which is mentioned in the revised manuscript in line 53-54: “For example, stem-binding antibodies CR6261^{5,6} and CR8020⁷ are in clinical trials (NCT02371668 and NCT01938352) as antivirals.”

- Lines 115-116: it would be worth mentioning that the breadth is characterized by the neutralization of several (not all) strains tested within each subtype. In addition, it would be worth mentioning that only a few (and amongst those being C05 the most important example) of the RBS antibodies are endowed with heterosubtypic activity (i.e. they are broad but within a specific subtype).

Response: Thank you for the comment. In the revised manuscript, we have improved the description of the neutralization breadth of C05 as suggested in line 119-122:

“Notably, one such antibody, C05, uses a single loop on heavy-chain complementarity-determining region 3 (CDR H3) to target HA RBS and is able to neutralize several, although not all, strains within pandemic subtypes from both group 1 and group 2 influenza viruses²¹.”

In the revised manuscript, we also mention that some RBS-targeted bnAbs are subtype-specific in line 61-64: “While some HA RBS-targeted bnAbs, including 8M2^{26, 27}, CH65²⁸ and 5J8^{29, 30}, are subtype-specific due to the higher sequence variability in the RBS and its proximal regions as compared to the stem region, others, including C05²¹, S139/1^{22, 23}, F045-092^{24, 25}, and 2G1^{26, 27}, display heterosubtypic activity”.

- A general criticism arises from the lack of analysis on the role of somatic mutations outside C05 HCDR3. In particular, in Ekiert et al. Nature 2012 it was shown that C05 carries a five-AA insertion in HCDR1 and that, in spite of making only minor contacts with HA, this insertion plays a role in the recognition with high affinity of contemporary H3 isolates. In addition, the highly mutated HCDR2 was shown to have important contact sites with the backside of HCDR3, with a possible stabilizing role. It would be important to evaluate the role of somatic mutations events (including the 5-AA insertion) in combination with the in vitro optimized HCDR3 variants for instance by testing these new variants in combination with the germlined backbone or with C05 variants selectively germlined in HCDR1 or HCDR2. This analysis would allow to put in a more biological context the analysis of C05 evolution, that is here just limited to an in vitro affinity maturation exercise. The understanding of the importance of the somatic insertion in HCDR1 is important for instance for a better understanding of the key evolution events required for the development of universal influenza vaccines targeting RBD.

Response: We agree that mutating CDR H1, CDR H2, and the rest of CDR H3 may help optimize or further understand the binding affinity and breadth of C05. However, we believe that investigating those structural elements by itself is a separate study. In the revised manuscript, we have acknowledged the role of other structural features in binding to HA and in optimizing C05 in line 374-378: “While our results provide valuable insight into the interaction between a peptide scaffold and HA RBS, complete optimization of C05 affinity and breadth would likely also involve to some extent CDR H1 and the rest of CDR H3, which makes additional contacts with neighboring regions of HA RBS that are less evolutionarily conserved²¹, as well as CDR H2, which may be important for stabilizing CDR H3²¹.”

- In the discussion (lines 363-365) it is suggested that antibody evolution and increased affinity is combined with a progressively reduced breadth. While the Authors are not tempted to generalize, it might be important mentioning other examples related to anti-stem antibodies, whereby affinity maturation is characterized by both increased breadth and potency (e.g. group 1 anti-stem antibodies evolving into group 1+2).

Response: We are indeed not attempting to generalize the tradeoff between affinity and breadth for all bnAbs. In the revised manuscript, we have explicitly stated that affinity maturation is accompanied by an increase in breadth for certain stem-binding antibodies in line 370-372: “However, this trade-off between affinity and breadth is unlikely to be universal to all bnAbs, as the affinity maturation of certain stem-binding bnAbs is characterized by both increased breadth and potency^{8, 11, 14}”.

Minor points:

- It is not always clear if the 3 round of panning have been made with the 3 viruses (H1, H3, H5). In Fig. 1 for example this detail should be mentioned, likewise in line 146.

Response: Figure 1 has been modified to include this point. The main text in line 154-155 is also modified accordingly: "For each of H1, H3, and H5 HAs, the screening was composed of three rounds of selection."

- Figure 5 legend: in the description of (e) panel the name of the corresponding virus is missing (i.e. A/Aichi/2/68)

Response: The description of panel e in the Figure 5 legend has been fixed.

- Suppl. Fig. 9: the legend should indicate that sialic acid is depicted in yellow

Response: Supplementary Figure 9 legend has been modified accordingly.

Reviewer #3 (Remarks to the Author):

Wilson and coworkers investigated a broadly neutralizing antibody that is directed against the receptor binding site of influenza hemagglutinin. They randomized the tip of the CDR3 paratope, generated a yeast display library and screened for binders with enhanced affinity to H1 HA or H3 HA, respectively. Statistical analysis of next generation sequencing data after screening round 3 revealed various types of variants. The authors generated a cocrystal of H1 HA with an affinity improved variant and thereby gained insight into subtype specificity. The data are technically sound and the combination of YSD screening and NGS provided a wealth of information on sequence preference with respect to subtype binding. While the obtained variants are of no direct value for antiinfective therapy, they provide valuable insight into evolutionary drift of subtype HA RBS. I can recommend the paper for publication without changes.

Response: Thanks for the encouraging comments.

REVIEWERS' COMMENTS:

Reviewer #2 (Remarks to the Author):

Authors have fulfilled the most critical issues that were raised.